# Fluorescence In Situ Hybridization (FISH) for the Characterization and Monitoring of Primary Cultures from Human Tumors

Ruth Román-Lladó [1,*], Cristina Aguado [1], Núria Jordana-Ariza [1], Jaume Roca-Arias [1], Sonia Rodríguez [1], Erika Aldeguer [1], Mónica Garzón-Ibañez [1], Beatriz García-Peláez [1], Marta Vives-Usano [1], Ana Giménez-Capitán [1], Andrés Aguilar [2], Alejandro Martinez-Bueno [2], María Gonzalez Cao [2], Florencia García-Casabal [3], Santiago Viteri [4], Clara Mayo de las Casas [1], Rafael Rosell [2] and Miguel Angel Molina-Vila [1]

1    Laboratory of Oncology, Pangaea Oncology, Dexeus University Hospital, 08028 Barcelona, Spain
2    Dr. Rosell Oncology Institute, Dexeus University Hospital, 08028 Barcelona, Spain
3    Instituto Oncológico Dr. Rosell, Centro Médico Teknon, QuirónSalud, 08028 Barcelona, Spain
4    UOMI Cancer Center, Clínica Mi Tres Torres, 08017 Barcelona, Spain
*    Correspondence: rroman@panoncology.com

**Abstract:** Genetic and drug sensitivity assays on primary cultures are not only of basic but also of translational interest and could eventually aid oncologists in the selection of treatments. However, cancer cells need to be identified and differentiated from the non-tumor cells always present in primary cultures. Also, successive passages can change the proportions of these two subpopulations. In this study, we propose fluorescence in situ hybridization (FISH) analysis on cell smears to determine the presence of tumor cells in primary cultures obtained from patients carrying translocations or copy number gains. FISH proved to be an easy, fast, economic, and reliable method of characterizing cell populations, which could be used repeatedly at different passages to monitor variations and to confirm the maintenance of translocations and copy number gains throughout the culture process.

**Keywords:** primary culture; FISH; malignant effusions; malignant ascites

## 1. Introduction

Primary cultures are defined as those directly obtained from fresh tissues. In the case of human tumors, primary cultures in 2D and 3D formats (including patient-derived organoids (PDOs) and patient-derived xenografts (PDXs)) are increasingly being used not only as research tools but also for practical purposes [1–3]. Clinical responses to antitumor treatments are heterogeneous and can be difficult to predict based on patients' genotypes; drug testing in cultures derived from solids or effusions can more accurately reflect the sensitivity profile of a tumor. For this reason, primary cultures have been used as preclinical models in the development of targeted agents [3–5] and are currently starting to be introduced in the clinic as a tool that could eventually help clinicians select the most appropriate treatments, complementing the histopathological and genetic analysis of the tumor [6,7]. However, their use is still anecdotal, and large observational studies are pending to correlate in vitro sensitivity with clinical outcomes [8–10].

Culturing tumor cells directly obtained from biopsies presents several difficulties. First, patients with metastatic disease are often diagnosed using not surgical but needle biopsies, where the tumor material is often scarce. In such scenarios, pleural effusions and ascites, where present, represent a valid alternative since relatively large numbers of viable cells representing the whole tumor heterogeneity can be isolated from such effusions [11,12]. Second, the success rate when trying to establish primary cultures is relatively low, ranging between 10 and 40% depending on the type of tumor and starting material [7]. Third, many primary cultures are plagued by the overgrowth of non-cancer cells, such

as stromal fibroblasts, leading to the quick disappearance of the tumor cells. Therefore, primary cultures should be regularly monitored to confirm that tumor cells are still present and retain the representative genetic alterations of the tumor of origin. Several methods have been used to characterize primary cell cultures, depending on which type of alteration needs to be assessed. These methods range from simpler procedures, such as immunocytochemistry, polymerase chain reaction (PCR), or the pathological examination of stained cells, to more complex assays that require larger amounts of starting material, such as gene expression analysis, flow cytometry, or next-generation sequencing (NGS) [7,9,11–13]. Characterization is routinely performed on cells after two to three weeks of culture and a few passages, especially when costly assays are used. Regarding fluorescence in situ hybridization (FISH), it has occasionally been employed in primary cultures of hematological malignancies for diagnostic purposes when the starting material obtained from the patient was insufficient [14,15]. However, despite being a quick, inexpensive, easy technique requiring small amounts of material, FISH has never been reported as a method of monitoring primary cultures derived from cancer patients.

In this article, we describe the utility of the FISH analysis on cell smears to characterize primary cultures presenting gene fusions or copy number gains (CNGs), assess the percentage of tumor cells, and follow the presence of genetic alterations in each passage.

## 2. Materials and Methods

### 2.1. Patient and Cell Line Samples

Ascitic and pleural fluids were obtained from the Dexeus University Hospital, Teknon Medical Center, and the UOMI Cancer Center with previous informed patient consent. The study was approved by the ethical committees of each hospital (approval number 04/2020, 25 February 2020) and was conducted according to the Declaration of Helsinki.

### 2.2. Primary Cultures

Ascitic or pleural fluids (10–500 mL) were kept in a sterile container at room temperature until processing (<24 h). Fluids were centrifuged at 2300 rpm for 10 min, and the cell pellet was resuspended in complete medium (CM) consisting of Roswell Park Memorial Institute (RPMI) medium supplemented with 10% fetal bovine serum (FBS) or human serum (HS), 50 mg/mL of penicillin-streptomycin, and 2 mM of L-glutamine. Cells were subsequently grown in T25 or T75 flasks in a humidified atmosphere with 5% $CO_2$ at 37 °C. In cases with abundant erythrocytes, the cell pellet was resuspended in 10 mL of CM, and the erythrocytes were removed via density gradient centrifugation in SepMate$^{TM}$ tubes with Lymphoprep$^{TM}$ (StemCell Technologies, Vancouver, BC, Canada) according to the manufacturer's instructions. If the initial material was sufficient, we cultured the cells from pleural effusion or ascites in parallel with FBS and HS. FBS is significantly cheaper and less variable than HS, but some primary cultures only grew in HS. In addition, in some cases, the medium was supplemented with 1 ng/mL of human hepatocyte growth factor (HGF) (Sigma-Aldrich, St Louis, MO, USA). The growth medium was replaced every 3–4 days. Cells were tested for mycoplasms every two weeks.

### 2.3. Cell Smear Preparation and FISH

Cells were obtained from a T25 flask at 50% confluence. After trypsinization, cells were pelleted by centrifugation at 1500 rpm for 5 min, washed with PBS, and fixed using 50–200 μL of Carnoy (3:1 methanol:acetic acid). Two to three droplets of fixed cells were placed in the center of a positively charged slide, avoiding overlapping; allowed to dry for at least six hours; and directly hybridized using the appropriate FISH probe and following the manufacturer's instructions. The probes to assess fusions were the ZytoLight$^®$ SPEC ALK Dual Color Break Apart Probe and ZytoLight$^®$ SPEC ROS1 Dual Color Break Apart Probe. CNGs were assessed using a ZytoLight$^®$ SPEC MYC Dual Color Break Apart Probe, a ZytoLight$^®$ SPEC MET/CEN 7 Dual Color Probe, a ZytoLight$^®$ SPEC FGFR1/CEN 8 Dual Color Probe, and a ZytoLight$^®$ SPEC EGFR/CEN 7 Dual Color Probe (all from

ZytoVision, Bremerhaven, FRG). The FISH probe locations and signal patterns are shown in Figure S1 and Table S1. For ALK and ROS1 fusion probes, a minimum of 50 cells were counted and the percentage of cells with break-apart signals was recorded (Table S2). For MET CNGs, two evaluation criteria were used: (i) the percentage of cells with a gene copy number $\geq 5$, which would indicate a CNG, and (ii) the percentage of cells with MET/CEN7 ratio (r) $\geq 2$, which would indicate that the CNG is a true amplification and not a consequence of other types of alterations leading to MET CNGs, such as polysomy (>2 copies of a single chromosome) or polyploidy (>2 copies of all chromosomes). CEN7 is a probe that binds to the chromosome 7 centromere. If the ratio of the MET gene vs. the chromosome 7 centromere is more than 2, it means that only the MET gene (and not the entire chromosome 7 or the entire genome) is amplified. Similarly, for FGFR1 and EGFR, a ratio $\geq 2$ and a gene copy number per cell $\geq 5$ were used. In the case of MYC, a gene copy number per cell $\geq 5$ was used as the positivity criterion for CNG. For the CNG assessment of EGFR, MYC, and FGFR1, a minimum of 30 cells were counted (Table S2).

### 2.4. NGS and nCounter

DNA was extracted using the DNeasy Blood & Tissue isolation kit (Qiagen, Hilden, FRG) according to the manufacturer's instructions, and concentrations were estimated using the Qubit 3.0 fluorometer (Invitrogen, Eugene, OR, USA). DNA-based NGS was performed with the GeneRead® QIAact Custom V2 DNA UMI Panel (Qiagen, Hilden, FRG), which can detect mutations in *EGFR*, *BRAF*, *MET*, *ERBB2*, *ALK*, *ROS1*, *RET*, *PIK3CA*, *KRAS*, *NRAS*, *KIT*, *PDGFRA*, *TP53 STK11*, *KEAP1*, *ARID1A*, *FAT1*, *NFE2L2*, *SETD2*, *POLE*, *POLD1*, *IDH1*, *IDH2*, *ERBB4*, *FGFR1*, *FGFR2*, and *FGFR3* and copy number variations (CNGs) in *EGFR*, *ERBB2*, *MET*, *FGFR1*, *FGFR2*, *FGFR3*, *BRAF*, *KRAS*, *MYC*, *CDK4*, and *CDK6*. Up to 40 ng of purified DNA were used as a template. Clonal amplification was performed on 625 pg of pooled libraries and, following bead enrichment, the GeneReader instrument was used for sequencing. Qiagen Clinical Insight Analyze (QCI-A) software was employed to align the read data and call sequence variants, which were imported into the Qiagen Clinical Insight Interpret (QCI-I) web interface for data interpretation and the generation of the final custom report. Copy number gains (CNGs) were calculated using an in-house algorithm, as described in [16,17]. Our NGS algorithm determines CNGs by comparing the coverage of a gene with the average coverage of all the genes in the NGS panel. Consequently, if a cell is polyploid, all the genes in the genome have multiple, equal numbers of copies, and our NGS pipeline cannot detect the increased CNGs, which affects the entire genome. In contrast, in the case of polysomy or true amplification, our algorithm will identify the CNGs in the target gene(s).

RNA was extracted using the High Pure RNA Isolation Kit (Roche, Basel, Switzerland) according to the manufacturer's instructions, and concentrations were estimated using the Qubit 3.0 fluorometer (Invitrogen, Eugene, OR, USA). RNA preparations were analyzed using an nCounter (NanoString Technologies, Seattle, WA, USA), as described in [18,19]. The total RNA was hybridized with a custom-designed mixture of biotinylated capture tags and fluorescently labeled reporter probes that included, among others, probes for 3′ regions of *ALK*, *ROS1*, and *RET* and probes for specific fusion transcripts. The codeset also included probes for housekeeping genes (*ACTB*, *PSMC4*, *GAPDH,* and *MRPL19*) and positive and negative controls. All processes of hybridization, capture, cleanup, and digital data acquisition were performed with the nCounter Prep Station® and Digital Analyzer®, according to the manufacturer's instructions. The reporter counts were collected with nSolver analysis software version 3.0 and analyzed as described in [18,19].

## 3. Results

### 3.1. Samples and Protocol

Thirty-seven pleural effusions and 19 ascitic fluids were collected from September 2020 to May 2022. Of them, samples from 12 patients (10 pleural effusions and 2 ascitic fluids) were positive for CNGs and/or fusions in a previous formalin-fixed, paraffin-embedded

(FFPE) biopsy, cytological extension, or in the cfDNA/RNA purified from the same fluid (Table S3). Most patients (9/12) had a diagnosis of lung adenocarcinoma; the remaining patients presented a non-small cell carcinoma (NSCLC) not otherwise specified, a high serous ovarian carcinoma, and a melanoma. Fusions in ALK and ROS1 or CNGs in MYC were detected in 2 samples each, EGFR and FGFR1 CNGs were detected in one sample each, and MET CNGs were detected in 6 samples. Among the 6 samples with MET CNGs, two corresponded to patients progressing to fusion-specific tyrosine kinase inhibitors and showed a concomitant ALK fusion (Table 1, Table S3).

**Table 1.** Types of fluids collected and known alterations in each sample.

| Sample | Type of Tumor | Collection Time | Fluid | Known Fusion/CNGs in Previous Material |
|---|---|---|---|---|
| 1 | Lung Adenocarcinoma | Progression | Pleural | *ALK* fusion/*MET* CNG |
| 2 | Lung Adenocarcinoma | Progression | Pleural | *ALK* fusion/*MET* CNG |
| 3 | High-Grade Serous Ovarian Carcinoma | Progression | Ascitic | *EGFR* CNG |
| 4 | Lung Adenocarcinoma | Basal | Pleural | *FGFR1* CNG |
| 5 | Lung Adenocarcinoma | Basal | Pleural | *MET* CNG |
| 6 | Lung Adenocarcinoma | Progression | Pleural | *MET* CNG |
| 7 | Melanoma | Progression | Ascitic | *MET* CNG |
| 8 | Lung Adenocarcinoma | Progression | Pleural | *MET* CNG |
| 9 | Lung Adenocarcinoma | Progression | Pleural | *MYC* CNG |
| 10 | NSCLC | Basal | Pleural | *MYC* CNG |
| 11 | Lung Adenocarcinoma | Progression | Pleural | *ROS1* fusion |
| 12 | Lung Adenocarcinoma | Progression | Pleural | *ROS1* fusion |

A protocol for fast, efficient FISH analysis in smears of cultured cells was developed based on the methodology usually employed for hematological samples [20]. It involved trypsinization of the cultures, fixation with carcinoid, extension on slides of a small number of cells (<1000), and staining with a standard FISH protocol. All primary cultures were successfully analyzed using this method. The turnaround time was 24 h; the quality of the images was comparable or superior to those obtained from FFPE tissues, with less background due to tissue overlapping (Figure 1); and the smears could be easily evaluated by an expert pathologist.

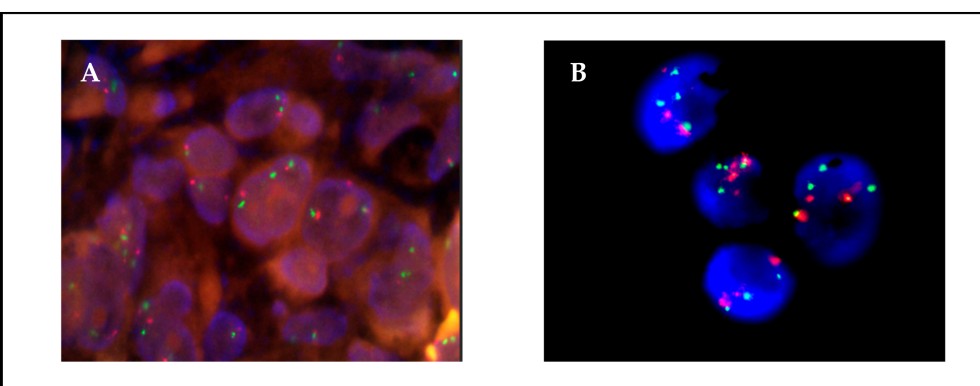

**Figure 1.** FISH *MET* images on (**A**) FFPE tissue and (**B**) a paired cell culture from patient 12. Since the patient was in progression, the possible presence of *MET* amplification was determined by FISH. Both tissue and culture samples were negative, with an average of 3 green signals corresponding to *MET*, a ratio of 1:1 to CEN7, and 0% of cells with >5 copies. Culture preparation presents less background and overlapping.

### 3.2. Detection of CNGs and Fusions in Low-Passage Primary Cultures

FISH analysis was performed in the 12 primary cultures at initial passages using the protocol described above. One of the primary cultures, corresponding to patient 7, did not harbor positive cells for *MET* CNGs, having a *MET/CEN7* ratio of ~1, and the number of copies was ~2 in all cells analyzed. In the 11 remaining primary cultures, cells positive for the same alteration(s) previously found in biopsies or cytologies were observed: either *ALK/ROS1* fusions or *MET/FGFR1/MYC* CNGs (Tables 2 and 3, Table S2). Depending on the primary culture and the passage, the percentage of cells harboring the expected alteration(s) ranged from 0 to 100%. Two primary cultures (samples 6 and 12) presented two cell populations, floating and adherent, which were analyzed separately by FISH. In both cases, the floating fraction presented a higher percentage of positive cells and was separated from the adherent cells and selected for subsequent culture (Figure 2). Also in both cases, the floating cells were alive and actively dividing. Remarkably, the floating cells were already present in the initial pleural effusions of the patients and were, therefore, adapted to grow in suspension. Although purely epithelial cells, such as adenocarcinoma cells, are not expected to float, the well-known phenomenon of epithelial-to-mesenchymal transition could have contributed to this adaptation.

**Table 2.** Results of detection of *ALK* and *ROS1* fusions by FISH in primary cultures. AC, adherent cells; FC, floating cells.

| Sample | Passage | Alteration | % Translocated Cells |
|---|---|---|---|
| Sample 1 | 2 | *ALK* fusion | 50 |
| Sample 1 | 3 | *ALK* fusion | 17 |
| Sample 1 | 5 | *ALK* fusion | 50 |
| Sample 1 | 8 | *ALK* fusion | 90 |
| Sample 1 | 9 | *ALK* fusion | 75 |
| Sample 1 | 11 | *ALK* fusion | 65 |
| Sample 1 | 13 | *ALK* fusion | 50 |
| Sample 2 | 2 | *ALK* fusion | 50 |
| Sample 2 | 3 | *ALK* fusion | 72 |
| Sample 2 | 5 | *ALK* fusion | 37 |
| Sample 11 | 1 | *ROS1* fusion | 9 |
| Sample 11 | 2 | *ROS1* fusion | 87 |
| Sample 11 | 5 | *ROS1* fusion | 78 |
| Sample 11 | 7 | *ROS1* fusion | 99 |
| Sample 12 AC | 1 | *ROS1* fusion | 76 |
| Sample 12 AC | 2 | *ROS1* fusion | 99 |
| Sample 12 AC | 5 | *ROS1* fusion | 100 |
| Sample 12 FC | 1 | *ROS1* fusion | 100 |
| Sample 12 FC | 2 | *ROS1* fusion | 100 |
| Sample 12 FC | 5 | *ROS1* fusion | 100 |

**Table 3.** Results of the detection of CNGs by FISH in primary cultures. Ratio refers to the number of copies of the corresponding gene per cell divided by the number of copies of the centromere where the gene is located (*MET*/CEN7, *EGFR*/CEN7, and *FGFR1*/CEN8, respectively). Copies refer to the average copy number of *MET*, *EGFR*, *FGFR1*, or *MYC*, respectively. % Cells refers to the percentage of cells carrying 5 or more copies of the corresponding gene. AC, adherent cells; FC, floating cells.

| Sample | Passage | Alteration | Ratio (gen/cen) | Copies | % Cells with ≥5 Gene Copies |
|---|---|---|---|---|---|
| Sample 1 | 5 | *MET* CNG | 1.45 | 5.1 | 77 |
| Sample 1 | 8 | *MET* CNG | 1.3 | 4.4 | 50 |
| Sample 1 | 9 | *MET* CNG | 1.3 | 4.5 | 53 |
| Sample 1 | 11 | *MET* CNG | 1 | 3.2 | 27 |
| Sample 1 | 13 | *MET* CNG | 1 | 3.0 | 5 |

**Table 3.** *Cont.*

| Sample | Passage | Alteration | Ratio (gen/cen) | Copies | % Cells with ≥5 Gene Copies |
|---|---|---|---|---|---|
| Sample 2 | 2 | *MET* CNG | 2.4 | 7.3 | 27 |
| Sample 2 | 3 | *MET* CNG | 1.2 | 3.5 | 7 |
| Sample 2 | 5 | *MET* CNG | 1.3 | 4.1 | 10 |
| Sample 5 | 2 | *MET* CNG | 0.9 | 10.2 | 100 |
| Sample 5 | 3 | *MET* CNG | 0.9 | 8.4 | 100 |
| Sample 6 AC | 1 | *MET* CNG | 8.5 | 17.1 | 36 |
| Sample 6 AC | 2 | *MET* CNG | 2.8 | 5.5 | 20 |
| Sample 6 FC | 1 | *MET* CNG | >10 | >20 | 81 |
| Sample 6 FC | 2 | *MET* CNG | >10 | >20 | 96 |
| Sample 6 FC | 8 | *MET* CNG | >10 | >20 | 100 |
| Sample 7 | 1 | *MET* CNG | 1 | 2 | 0 |
| Sample 8 | 1 | *MET* CNG | 1 | 3.5 | 20 |
| Sample 8 | 3 | *MET* CNG | 1 | 2 | 0 |
| Sample 3 | 1 | *EGFR* CNG | 1 | 6.7 | 100 |
| Sample 3 | 3 | *EGFR* CNG | 1 | 6.2 | 90 |
| Sample 4 | 1 | *FGFR1* CNG | 3.6 | 15.1 | 67 |
| Sample 4 | 3 | *FGFR1* CNG | 3 | 11.2 | 70 |
| Sample 9 | 1 | *MYC* CNG | - | >6 | 72 |
| Sample 9 | 2 | *MYC* CNG | - | >6 | 63 |
| Sample 9 | 6 | *MYC* CNG | - | 7.8 | 90 |
| Sample 10 | 2 | *MYC* CNG | - | >6 | 100 |
| Sample 10 | 6 | *MYC* CNG | - | >6 | 100 |

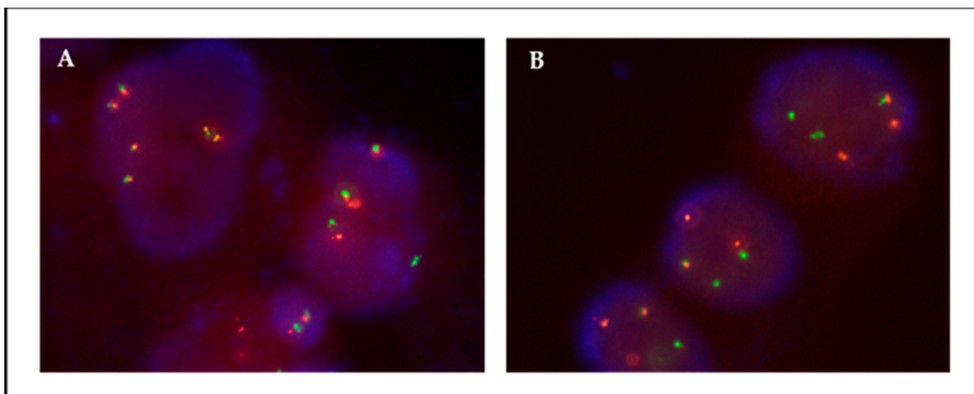

**Figure 2.** Patient 12, FISH *ROS1* on passage 1; (**A**) 76% of the adherent cells were translocated; (**B**) 100% of the floating cells showed translocations. Translocation is observed as isolated green and orange signals.

### 3.3. Monitoring of Tumor Cells in Primary Cultures by FISH

The eleven primary cultures positive, by FISH, for gene fusions (*n* = 4), CNGs (*n* = 7), or both (*n* = 2) were monitored in subsequent passages, and detailed FISH analysis at different passages allowed for the detection of variations in the proportion of the different cell populations. Positivity for *ALK* or *ROS1* fusions was maintained (Figure 3), and the percentage of cells carrying the translocations significantly increased in the case of patient 11, remaining more or less stable in the other three cases (Table 2). Regarding CNG alterations, *MET* CNGs sharply declined in three cases—patients 1, 2, and 8; however, it was maintained in two cases—patients 5 and 6 (Table 3). As an example, the monitoring of samples from patient 1 revealed a sharp decrease in the number of cells carrying ≥ 5 copies

(Figure 4A). Interestingly, when cells were cultured in parallel in the presence and absence of hepatocyte growth factor (HGF, the ligand of the MET receptor), *MET* CNGs were more frequent in the presence of HGF (Figure 5).

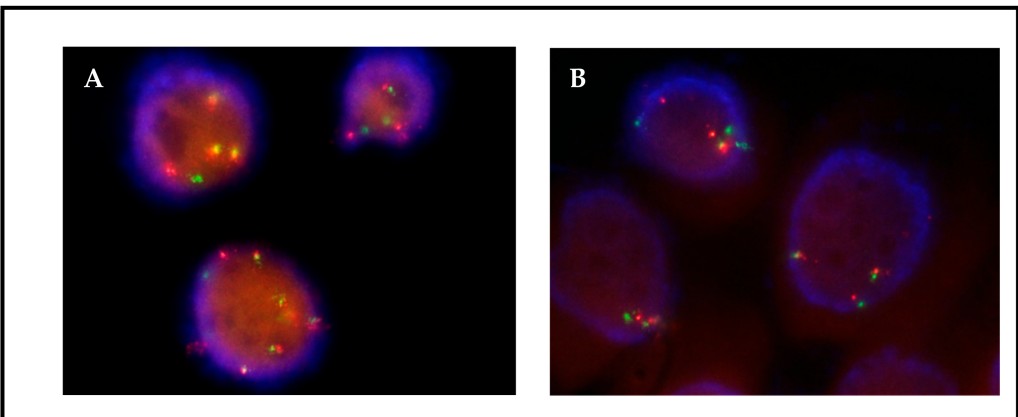

**Figure 3.** Patient 1, FISH *ALK* on passage 2 (**A**) and passage 13 (**B**). *ALK* translocation is maintained through culture. Translocation is observed as isolated green and orange signals.

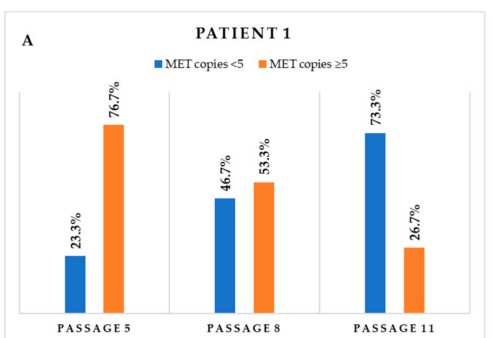
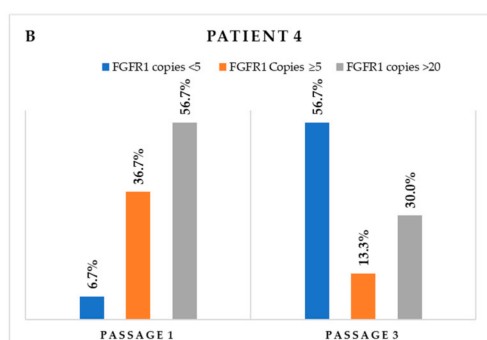

**Figure 4.** Monitoring of the variation through the different passages in cell populations carrying (**A**) *MET* CNGs and (**B**) *FGFR1* CNGs.

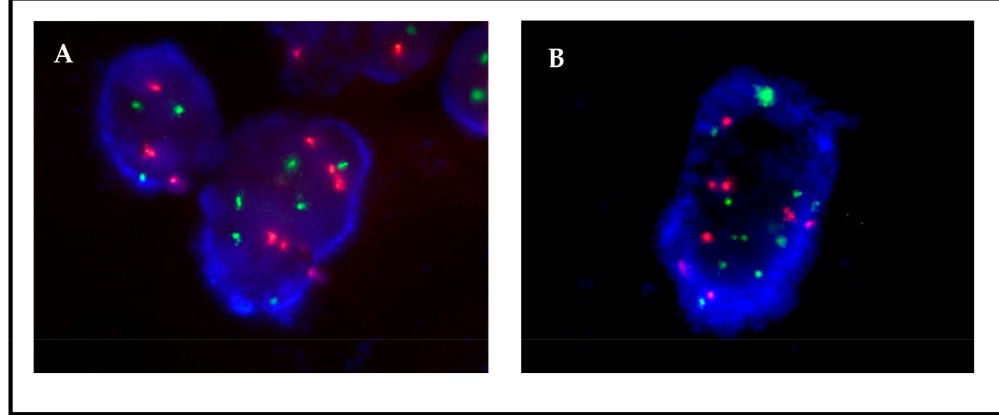

**Figure 5.** Patient 1, FISH *MET* on passage 13 (**A**) without HGF and (**B**) with HGF. *MET* gene copies are observed as green signals. A CNG pattern corresponding to ≥5 green signals is more frequently detected in the presence of HGF.

Regarding *MYC*, *FGFR1*, and *EGFR* CNGs, positivity was maintained in the three cases bearing these alterations (Table 3). Nonetheless, detailed FISH counts showed a decrease in the number of cells carrying ≥ 5 copies (Figure 4B).

### 3.4. Comparison with NGS and nCounter

Primary cultures were submitted to either next-generation sequencing (NGS) or nCounter to validate the results of the FISH analysis. The presence of *ALK* and *ROS1* fusion transcripts was confirmed by nCounter in the four samples carrying these alterations. Similarly, the presence or loss of CNG could also be corroborated by NGS in all cases; with only the exception of sample 5. This sample showed high polyploidy, with an average of 10 copies of *MET*, which was detected by FISH but not by NGS (Tables 4 and S4).

**Table 4.** Correlations between FISH and nCounter (upper rows) or NGS (lower rows) for the detection of fusions and CNGs. ND, not detected. Rows in bold indicate the type of items listed below.

| **FISH** | | | | | **nCounter** | |
|---|---|---|---|---|---|---|
| **Sample** | **Passage** | **Alteration** | **Ratio** | **Copies** | **% Positive Cells** | **Fusion** | **Exons** |
| Sample 1 | 3 | *ALK* | - | - | 17 | *EML4-ALK* | v1 (E13:A20) |
| Sample 1 | 8 | *ALK* | - | - | 90 | *EML4-ALK* | v1 (E13:A20) |
| Sample 1 | 9 | *ALK* | - | - | 75 | *EML4-ALK* | v1 (E13:A20) |
| Sample 1 | 13 | *ALK* | - | - | 50 | *EML4-ALK* | v1 (E13:A20) |
| Sample 2 | 2 | *ALK* | - | - | 50 | *EML4-ALK* | v1 (E13:A20) |
| Sample 2 | 3 | *ALK* | - | - | 72 | *EML4-ALK* | v1 (E13:A20) |
| Sample 11 | 2 | *ROS1* | - | - | 87 | *ROS1* | not identified * |
| Sample 11 | 5 | *ROS1* | - | - | 78 | *ROS1* | not identified * |
| Sample 11 | 7 | *ROS1* | - | - | 99 | *ROS1* | not identified * |
| Sample 12 AC | 1 | *ROS1* | - | - | 76 | *CD74-ROS1* | C6-E34 |
| Sample 12 FC | 1 | *ROS1* | - | - | 100 | *CD74-ROS1* | C6-E34 |
| Sample 12 FC | 24 | *ROS1* | - | - | 100 | *CD74-ROS1* | C6-E34 |
| **Sample** | **Passage** | **Alteration** | **Ratio** | **Copies** | **% Cells with ≥5** | **NGS** | |
| Sample 1 | 5 | *MET* CNG | 1.45 | 5.1 | 77 | *MET* | 10 copies |
| Sample 1 | 11 | *MET* CNG | 1 | 3.2 | 27 | ND | ND |
| Sample 2 | 2 | *MET* CNG | 2.4 | 7.3 | 25 | *MET* | 15 copies |
| Sample 3 | 1 | *EGFR* CNG | 1 | 6.7 | 100 | *EGFR* | 5 copies |
| Sample 4 | 1 | *FGFR1* CNG | 3.6 | 15.1 | 90 | *FGFR1* | 16 copies |
| Sample 5 | 2 | *MET* CNG | 0.9 | 10.2 | 100 | ND | ND |
| Sample 6 FC | 1 | *MET* CNG | >10 | >20 | 81 | *MET* | >50 copies |
| Sample 7 | 1 | *MET* CNG | 1 | 2 | 100 | ND | ND |
| Sample 8 | 1 | *MET* CNG | 1 | 3,5 | 0 | ND | ND |
| Sample 9 | 2 | *MYC* CNG | - | >6 | 63 | *MYC* | 7 copies |
| Sample 10 | 2 | *MYC* CNG | - | >6 | 100 | *MYC* | >50 copies |

* The *ROS1* rearrangement in this patient was detected by FISH, a technique that does not identify the partner. To this end, we submitted the sample to nCounter. However, it did not show any signal for any of the partner-specific probes in the nCounter panel, and only showed a signal for the "common" probes for all *ROS1* rearrangements [19]. Consequently, we could not identify the partner.

## 4. Discussion

Drug sensitivity assays on primary cultures and PDXs are increasingly used for research purposes since they are closer to the real tumor than established cell lines. Some pilot studies have also indicated that they can eventually aid oncologists in the selection of treatments [21], although large observational studies are pending to correlate in vitro sensitivity with clinical outcomes [8–10]. In addition, in patients with biopsies or cytological samples harboring an insufficient number of malignant cells, primary cultures could provide enough cells for genetic testing. Malignant effusions, which typically occur in advanced-stage patients after treatment failure, are particularly suited for the establishment of primary cultures and PDXs since they often contain few stromal components and large

numbers of tumor cells already adapted to grow in a liquid environment [11,22,23]. In this study, we collected 12 malignant effusions from melanoma (*n* = 1) and lung (*n* = 10) and ovarian (*n* = 1) carcinomas, and we could establish primary cultures from 9/10 lung adenocarcinomas and the ovarian tumor. In contrast, tumor cells did not grow in the case of the melanoma sample. Although just one case is not sufficient to draw any definite conclusions, melanoma cells might be more difficult to grow and require a special protocol.

In this scenario, one of the first issues to be addressed was whether effusions contain viable tumor cells. Here, we demonstrated that conducting FISH analysis on a minimal amount of material collected after centrifugation or from the first culture passages can easily confirm the presence of tumor cells carrying fusions or CNGs in less than 24 h. NGS and whole genome sequencing have also been used to analyze tumor cell primary cultures [24–27]. These high throughput technologies undoubtedly provide much more information than simply the presence or absence of a single alteration; however, they are more appropriate for the full characterization of well-established cell cultures and less suited to be used in early passages to confirm the presence of tumor cells. In particular, in contrast to FISH, they require a large amount of material, are time-consuming, and are far more expensive. Our study also proved an excellent correlation between FISH results and those obtained with NGS or nCounter. Moreover, FISH allowed for the detection of polyploidy and CNGs in a small percentage of cells that were not detected by NGS [17]. Finally, in primary cultures showing adherent and floating cells, FISH also allowed for the fast characterization of the subpopulations and the selection of those harboring a higher proportion of tumor cells.

In addition, FISH can be repeatedly used during the primary culture to monitor variations in the proportion of tumor cells over time while allowing for culture conditions to be modified, if needed. The proportion of cells carrying driver alterations in tumor biopsies is not considered when determining patient treatments. The reason is that drivers (such as *EGFR*, *KRAS*, or *BRAF* mutations; *ALK* or *ROS1* fusions; baseline CNGs; etc.) are clonal, meaning that they are present in all malignant cells. However, when establishing a primary culture, cancer cells might be lost and replaced by stromal cells, making the culture unsuitable for genetic analyses or drug testing. The percentage of tumor cells in primary cultures can be determined by testing for the driver originally present in the tumor since stromal cells lack it. Consequently, in the case of fusion or CNG-positive tumors, FISH analysis can be adapted to determine the abundance of malignant cells in a primary tumor in a fast and efficient way, as we have demonstrated in this study. Based on FISH, we determined that the percentage of malignant cells in our primary cultures ranged from 0 to 100%, as shown in Tables 2 and 3. Also using FISH, we were able to monitor the evolution of our primary cultures, which was extremely variable. In most cases, cells positive for the rearrangement were enriched (such as samples 6, 9, 11, or 12), in one case they were lost (sample 8) and, finally, some primaries showed oscillations in the percentage of positive cells (samples 1 and 2). In this respect, the enrichment in positive cells observed in most primary cultures could be a consequence of the fastest growth rate of the tumor cells and the programmed death of stromal cells after a limited number of passages.

In the case of acquired alterations at progression to therapies, such as *MET* amplification in tumors progressing to fusion-specific TKIs, the situation is different. Acquired alterations are subclonal and, therefore, not present in all tumor cells. Consequently, the changes in the percentage of positive cells in primary cultures might also reflect the selection of different subclones of the original tumor. In our study, two patients at progression presented the initial *ALK* fusion together with an acquired *MET* CNG. *MET* CNGs have been widely reported as a resistance mechanism to treatment with EGFR tyrosine kinase inhibitors and have recently been described as a resistance mechanism in 15% of tumor biopsies from patients relapsing on ALK inhibitors [28,29]. In contrast to *ALK* or *ROS1* fusions, acquired *MET* CNGs were not stable in culture, being maintained only in the first passages but sharply decreasing over prolonged culture. For instance, in the case of sample 1, *ALK*-positive cells were stable but *MET*-amplified cells were completely lost—an

evolution that could be reverted by the addition of HGF (the MET ligand) to the culture. This finding strongly suggests that *MET* CNG only provides a competitive advantage to tumor cells in the presence of ligands able to activate the receptor. If this is not the case, cells grow more slowly than their *MET*-negative counterparts and are consequently lost.

**Supplementary Materials:** The following supporting information can be downloaded at: https: //www.mdpi.com/article/10.3390/jmp4010007/s1, Figure S1: (A) Design of ALK and ROS1 probes used for translocation studies and diagrams of the patterns observed in translocated and non-translocated interphase cells. (B) Design of MET/CEN7, EGFR/CEN7, FGFR1/CEN8 and MYC probes used for CNG assessment and diagrams of interphase cells with and without CNG patterns; Table S1: FISH probe designs and signal patterns. NA: True amplification can not be assessed with ZytoLight ® SPEC MYC Dual Color Break Apart; Table S2: Further details on percentage of cells with alterations and number of interphase cells counted for each sample passage; Table S3. Further details of the patients and samples included in the study; Table S4. Quality control (QC) parameters of the NGS analyses presented in Table 4.

**Author Contributions:** Conceptualization, R.R.-L., M.A.M.-V. and C.A.; methodology, R.R.-L., C.A., M.G.-I. and C.M.d.l.C.; formal analysis, R.R.-L., C.A., A.G.-C. and C.M.d.l.C.; investigation, R.R.-L., C.A., N.J.-A., J.R.-A., S.R., E.A., M.G.-I., B.G.-P., M.V.-U., A.G.-C. and C.M.d.l.C.; resources, A.A., A.M.-B., M.G.C., F.G.-C., S.V. and R.R.; data curation, R.R.-L. and C.A.; writing—original draft preparation, R.R.-L. and C.A.; writing—review and editing, all authors; visualization, B.G.-P. and S.R.; supervision, M.A.M.-V.; project administration, R.R.-L. and C.A.; funding acquisition, C.A. All authors have read and agreed to the published version of the manuscript.

**Funding:** This study was funded by the European Union Eurostars program (113736/22/36103/Ae).

**Institutional Review Board Statement:** The study was conducted in accordance with the Declaration of Helsinki, and approved by the ethical committees of each hospital (approval number 04/2020, 25 February 2020).

**Informed Consent Statement:** Informed consent was obtained from all subjects involved in the study.

**Data Availability Statement:** Data supporting the findings of this study are presented within the article and its Supplementary Data files, or available upon reasonable request from the corresponding authors.

**Conflicts of Interest:** The authors declare no conflict of interest.

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
