# Peer review of "Fluorescence In Situ Hybridization (FISH) for the Characterization and Monitoring of Primary Cultures from Human Tumors"

_jmp, doi:10.3390/jmp4010007_

Round 1

Reviewer 1 Report

The authors performed primary cell cultures of tumor cells and FISH analysis. A number of samples have been analyzed; however, the purpose of this study is not clear.

Major points

1.       Advantages of FISH have already been recognized in this field. It seems to me that comparisons of tumor cells between cell culture stages work as a part of quality control of cellular materials for in vitro experiments.

2.       As no explanation is given for FISH signal patterns and probe colors, FISH images cannot be understood. Please make a table which present FISH probes and normal signal patterns.

3.       Although ‘selection’ is emphasized in the title, this study performed single culture for each sample. Because of no replicates of primary cultures, selection of primary cultures cannot be considered.

4.       Please describe the number of interphase cells observed for analysis. The results should be fully provided in a supplementary data.

5.       Cell culture requires passaging by dilution of cells, different from in vivo tumor progression. Does monitoring of primary cultures help oncologist in the selection of treatments?

Minor points

1.         Please show FISH images of ALK and ROS1 signals.

2.         Each fusion should be described including positions of exons.

3.         Please provide QC data from NGS analysis, which includes mean coverage depth, percentage of targets coverage over 100 reads and targets under 20 reads.

Reviewer 2 Report

The manuscript entitled Fluorescence in situ hybridization (FISH) for the selection and monitoring of primary cultures of tumor cells by Roman-Llado et al. is interesting. I have several comments

Some sentences need to be corrected:

-       FISH and FFPE, NSCLC are directly written in abbreviation, authors have to write it in full the first time

-       Sometimes it is written CNG sometimes GNC, I don’t see the difference between copy number gain or gene copy number. If there is, it is confusing to have both abbreviation that are very similar. If there is no difference, please choose only one term.

-       Line 123: correct cytological

-       All genes need to be in italic (ALK, ROS1 ect…)

Introduction

Can authors explain why the proportion of cells carrying markers is critical for patient management? Clinician just check if the rearrangement or mutation is present to use TKI. At least for NSCLC

Materiel and Methods

The type of cancer is only written in the table 1 and was not obvious. Please make a sentence in this section explaining which type of cancer you work on. Do authors know if all the NSCLC were adenocarcinoma? Or NOS?

The percentage of malignant cells in the effusion is not indacted, I think it is interesting to have this information.

Are authors sure that the cells that are cultivated are only malignant cells? authors don’t show controls with carcinomatous marker as BerP4, EpCAM, or EMA for carcinoma and HMB45 or PS100 for melanoma.

Why did authors use FBS or HS for culture? Is there any difference?

Can authors explain what the ratio MET/CEP7 is for? How is it different or similar from the copy gain number that you can also see with FISH? Lines 87 -90 need to be more explained, in particular with the use of abbreviations CNG and GNC!

Result

Is the presence of ALK fusion and MET CNG already described in the literature? I think it is very rare (as ALK rearrangement is only present in 3-7% of NSCLC). It must be discussed in the discussion part. Were ALK fusion and MET CNG both present at diagnosis?

What technique was used to find the rearrangement in the effusion and before the culture? IHC? FISH? NGS? If it was FISH, I think it is interesting to know the percentage of cells that carry the mutation in the fluid before culture. And also, to know what method was used to find the rearrangement.

Lines 133-134: I am surprised that the images from cultured cell smears are not better than the FFPE. As in FFPE it’s section of cells whereas with cytology we can see the whole cell. Most of the time FISH is easier to read on cytology than on FFPE.

Concerning patient 7, authors did not discuss that it is the only sample from melanoma and maybe this type of cells is more difficult to grow? Did authors check that cells in culture were melanoma cells? Maybe it needs a different protocol to grow? Same for patient 3 with ovarian carcinoma.

Authors don’t discuss at all the % variation of cells carrying rearrangement between the passages. Do authors think that they selected clones? How can they explain it? Again, are all cells malignant in the culture?

In the legend of table 3 please explain ratio, copies and % cells with >5

 Concerning culture with floating cells, were the floating ones alive cells or were they dead? Because as these malignant cells are epithelial, they should not be floating… 

How can authors explain that the MET CNG in patient 5 can only be seen with FISH and not NGS?

In table 4: I don’t understand “non variant” for ROS1, what do authors mean? Is it rearranged or not? What is v1? For ALK and ROS1 it is not mutation but translocation/rearrangement.

Please explain abbreviations ND and NA in a table legend. Authors explain that there is only one patient (patient 5) where the NGS is different from FISH but why is it written ND and NA for sample 1/2/5/7 and 8? a column title is missing.

Why did authors add HGF for patient 1? Why this growth factor and not another one? Can authors cite an article that did the same?

The discussion needs to be modified

Results are not enough discussed

The use of culture for treatment selection is anecdotic, even if this manuscript is more for researcher than clinician, this sentence must be changed, or other articles must be cited.

Lines 208-209 “variations in the proportion of tumor cells over time”, does it mean there is other type of cells in culture? If yest it must be explained, if authors mean “tumor cells with rearrangement/amplification” please rephrase.

Authors show that FISH is feasible on malignant cell culture, but do they think it can be part of the decision treatment or do they see it to test drugs or better understand these mutations/fusions?

Round 2

Reviewer 2 Report

authors responded to comments and modified the manuscript as asked.

Several minors comments:

Line 27 : abbreviations for PDO and PDX

Lines 30 and 41: change "liquid biopsies" for effusions. Liquid biopsie is used for the search of molecular alterations by NGS on blood.

3.1. Samples and protocol: use italic for gene names.

Author Response

Reviewer #2

  1. Line 27: abbreviations for PDO and PDX

    -We have written in full the meaning of the abbreviations

  1. Lines 30 and 41 Change “liquid biopsies” for effusions. Liquid biopsie is used for the search of molecular alterations by NGS on blood.

-Corrected

  1. Samples and protocol: use italic for gene names

    -Corrected